# Molecular Features Accompanying Richter’s Transformation in Patients with Chronic Lymphocytic Leukemia

**DOI:** 10.3390/ijms26125563

**Published:** 2025-06-10

**Authors:** Xiaole Wang, Jingyu Chen

**Affiliations:** Department of Biochemistry and Molecular Biology, West China School of Basic Medical Sciences & Forensic Medicine, Sichuan University, Chengdu 610041, China

**Keywords:** molecular mechanisms, Richter’s Transformation, chronic lymphocytic leukemia, genetics, signal pathways, immune microenvironment, epigenetics

## Abstract

Chronic Lymphocytic Leukemia (CLL) is a highly heterogeneous tumor. Although targeted therapies such as Bruton’s Tyrosine Kinase (BTK) inhibitors and B-cell lymphoma-2 (Bcl-2) inhibitors have significantly improved patient outcomes in CLL, the disease remains incurable. A critical aspect of CLL progression is its transformation from an indolent tumor to a high-grade malignancy, a process known as Richter’s Transformation (RT) or Richter Syndrome. Treatment options for RT are very limited, and patient prognosis is often poor. The molecular mechanisms driving RT are not yet fully elucidated. This review aims to summarize recent advances in research aimed at uncovering the mechanisms underlying RT in CLL. By integrating findings from genetics, signaling pathways, epigenetics, and the tumor microenvironment, this review seeks to provide insights that could guide further basic research into RT and inform the development of novel therapeutic strategies to improve patient outcomes.

## 1. Introduction

Chronic Lymphocytic Leukemia (CLL) is a clonal proliferative disorder of mature B lymphocytes, characterized clinically by increased lymphocyte counts in the peripheral blood, hepatosplenomegaly, and lymphadenopathy [1,2]. CLL typically follows an indolent course, and advancements in treatment over the past few decades have significantly improved patient outcomes, with a five-year survival rate now approaching 90% [3,4]. However, approximately 2–9% of CLL patients undergo Richter’s Transformation (CLL-RT), a form of clonal evolution characterized by aggressive histological progression [5]. The molecular complexity of RT and its invasive development pose significant challenges to treatment, resulting in markedly reduced CLL patient survival. The median overall survival for CLL-RT patients is typically less than 12 months [5,6]. While RT can manifest as Hodgkin Lymphoma (HL) in a smaller subset of cases, or rarely as Interdigitating Dendritic Cell Sarcoma, Lymphoblastic Lymphoma, or T-Cell Lymphoma, the majority of cases (90–95%) transform into Diffuse Large B-Cell Lymphoma (RT-DLBCL) [5,7,8,9]. The analysis of *IGHV* gene rearrangements in RT-DLBCL shows that approximately 80% of RT-DLBCL patients have clones originating from the CLL phase, while the remaining 20% exhibit *IGHV* different from the CLL phase, suggesting a new onset of DLBCL which often has a better prognosis, with overall survival (OS) extending to 60 months [10,11,12]. This review will primarily focus on the molecular mechanisms involved in CLL clonally related RT-DLBCL, the most common and clinically significant subtype of CLL-RT.

Current therapeutic strategies for RT are largely tailored to the type of transformation. For RT-DLBCL, first-line treatment typically involves immunochemotherapy regimens applied in DLBCL, such as R-CHOP (rituximab, cyclophosphamide, doxorubicin, vincristine, and prednisone), O-CHOP (ofatumumab, cyclophosphamide, doxorubicin, vincristine, and prednisone), and R-EPOCH (rituximab, etoposide, prednisone, vincristine, cyclophosphamide, and doxorubicin) [13]. Despite these interventions, treatment efficacy remains suboptimal, with median OS ranging from 5 to 10 months. Furthermore, these regimens are often associated with severe toxicities, which are particularly burdensome for RT patients who are frequently frail and have compromised organ function [14,15,16].

The advent of targeted therapies has revolutionized CLL treatment, significantly improving patient prognosis and reducing the incidence of RT [17]. It has been demonstrated that covalent Bruton’s Tyrosine Kinase (BTK) inhibitors reduced the CLL-RT transformation rate to 2% [18]. Beyond BTK inhibitors, B-cell lymphoma-2 (Bcl-2) inhibitors and PI3K inhibitors, either as monotherapies or in combination, have emerged as promising therapeutic strategies for preventing CLL-RT [19,20]. However, for CLL patients who have already received targeted therapies or developed resistance to them, particularly in light of the rising prevalence of targeted therapy resistance, the benefits of small-molecule targeted therapies in treating or preventing RT are very limited [21].

Given these challenges, a deeper understanding of the fundamental biological mechanisms driving RT is essential for the development of novel therapeutic approaches and the optimization of treatment strategies. This review aims to synthesize recent advances in basic research on RT, especially RT-DLBCL, with a focus on genetic alterations, dysregulated signaling pathways, epigenetic modifications, and the role of the tumor microenvironment. By elucidating these mechanisms, we hope to provide insights that will shed light on the future directions in understanding the molecular pathogenesis in RT and the development of efficient combination treatment or novel targeted therapies for the disease.

## 2. Genetic Alterations

Key genetic mutations associated with RT include *TP53* (Cellular tumor antigen p53) deletion or mutation, *CDKN2A/B* (Cyclin-dependent kinase inhibitor 2A/B) deletion, *NOTCH1* (Neurogenic Locus Notch Homolog Protein 1) mutation, and *MYC* (v-myc avian myelocytomatosis viral oncogene homolog) gene amplification or overactivation due to chromosomal translocations (Figure 1). Notably, mutations in *TP53*, *CDKN2A/B*, *NOTCH1*, and *MYC* frequently co-occur in RT [10,13,22,23], highlighting their critical role in the pathogenesis of RT and underscoring the importance of elucidating their mechanisms to advance our understanding of this aggressive transformation.

### 2.1. TP53

In CLL, disruptions in *TP53*, such as 17p (Chromosome 17 short arm) deletion or mutations, are clear markers of poor prognosis. Approximately 60% of RT patients exhibit *TP53* disruption, and notably, over 50% of RT patients have *TP53* mutations present in the original CLL clone [24]. Although *TP53* mutations are well-established genetic alterations associated with RT, the precise mechanisms by which these mutations drive disease progression and sustain the malignant phenotype remain poorly understood. *TP53* mutations likely cooperate with other molecular factors to promote RT. For instance, an increased baseline accessibility of Transcription Factor 3 (TCF3) transcription factor motifs has been observed in the lymph nodes (LNs) and peripheral blood mononuclear cells (PBMCs) of RT patients with *TP53* deletion, suggesting a potential role for TCF3 in modulating the transcriptional landscape of *TP53*-mutated RT [25]. Exportin 1 (XPO1) is a nuclear export protein that plays a critical role in TP53 protein nuclear–cytoplasmic shuttling and maintaining cellular homeostasis. Selinexor, a selective XPO1 inhibitor, has demonstrated clinical activity in 40% of RT-DLBCL patients who were refractory to prior chemotherapy regimens (two out of five patients achieved partial responses), highlighting XPO1 as a potential therapeutic target in *TP53*-mutated RT [26]. Furthermore, Ataxia telangiectasia-mutated gene (ATM) serves as an upstream regulator of TP53 and is a key mediator of the DNA damage response pathway. Mutations in *ATM* frequently co-occur with *TP53* mutations or deletions [27], suggesting a synergistic role in driving genomic instability and disease progression in RT.

### 2.2. CDKN2A/B

*CDKN2A* and *CDKN2B* are critical tumor suppressor genes located at the 9p21 (Chromosome 9, short arm, position 21) locus, playing essential roles in cell cycle regulation by encoding the p16INK4A and p15INK4B proteins, which inhibit cyclin-dependent kinases (CDKs) and prevent cell cycle progression [28,29,30]. Homozygous deletions of *CDKN2A/B* are frequently observed in RT, with approximately 30% of RT patients exhibiting these deletions [13,22]. These genetic alterations are often associated with the inactivation of TP53 and frequently co-occur with the activation of MYC, a potent driver of cell proliferation and oncogenesis [31]. A retrospective study of 653 patient samples demonstrated that both *TP53* loss and *CDKN2A/B* deletions are independently significant predictors of poor prognosis in RT, underscoring their critical roles in disease progression [32,33]. In RT cells harboring deletions of *TP53* and *CDKN2A/B*, the activation of the B-cell receptor (BCR) signaling pathway further exacerbates cell proliferation and survival. Importantly, the inhibition of the BCR pathway in these cells almost completely suppresses their growth, indicating that RT tumors with these genetic alterations remain highly dependent on BCR signaling for their survival. This dependency highlights the potential therapeutic value of targeting the BCR pathway in RT patients with *CDKN2A/B* and *TP53* deletions [34,35].

### 2.3. MYC

MYC and NOTCH1 are both critical regulators of cell proliferation and play pivotal roles in the pathogenesis of RT. *MYC* mutations are detected in 30–50% of RT patients, and aberrant MYC activity is a major driver of the aggressive phenotype observed in B-cell lymphomas [36,37,38]. Notably, *MYC* mutations frequently coexist with *TP53* mutations in RT, suggesting a synergistic role in promoting genomic instability and disease progression [10]. *MYC* upregulation exerts broad effects on cellular processes, including significant alterations in the expression and function of Interferon Regulatory Factor 4 (IRF4), a transcription factor essential for B-cell differentiation and survival. These changes in IRF4 may contribute to the dysregulation of immune responses and the survival advantage of RT cells [39]. Max Gene Associated (MGA), a functional suppressor of MYC, plays a key role in regulating oxidative phosphorylation (OXPHOS) through its interaction with Nucleoside Diphosphate Kinase (NME1) [40,41]. As a tumor suppressor, MGA maintains metabolic homeostasis by repressing MYC-driven transcriptional programs. However, *MGA* deletion or inactivation disrupts this regulatory balance, leading to the upregulation of MYC-regulated and OXPHOS-related genes [40]. Recent studies have demonstrated that *MGA* deletion promotes the survival and proliferation of RT cells in preclinical models, highlighting MGA’s role as a critical regulator of MYC activity and the importance of the MGA-NME1-OXPHOS axis in driving RT progression [40].

### 2.4. NOTCH1

NOTCH1, a transmembrane receptor in CLL, releases its intracellular domain (NICD) upon ligand activation. *NOTCH1* mutations lead to NICD accumulation in the nucleus, prolonging signal transduction and promoting cell proliferation while inhibiting apoptosis, often resulting in aggressive disease transformation [42,43]. Gain-of-function *NOTCH1* mutations are found in approximately 30% of RT patients [31], and 45% of CLL patients with *NOTCH1* mutations develop RT [44]. NOTCH1 and the BCR pathways are functionally interconnected, with a NOTCH/PI3K/AKT axis identified in B-cell malignancies [45,46,47,48]. *NOTCH1* mutations are often observed in patients with trisomy 12 and frequently co-occur with *TP53* or *MYC* disruptions [23,31]. NICD accumulation in the nucleus leads to the transcriptional activation of multiple target genes, including *MYC* and genes involved in Nuclear factor kappa-B (NF-κB) signaling [43,49]. Additionally, intrinsic NOTCH1 signaling in RT is induced by CD4^+^ T cells expressing Delta-like protein 1 (Dll1) in the tumor microenvironment (TME), highlighting the importance role of the environmental immune cells in RT [50].

### 2.5. BCR-Related Gene Alterations

Gene alterations associated with the BCR pathway are closely linked with RT. Stereotyped immunoglobulin genes are common in clonally related RT-DLBCL (approximately 50%) [51]. Stereotyped BCRs lead to a significant activation of CLL cells (Figure 1), increasing the likelihood of RT [52]. For example, the stereotyped BCR subgroup 8 [53], which is characterized by the IGHV4-39/IGHD6-13/IGHJ5 (Immunoglobulin Heavy Variable 4–39/Immunoglobulin Heavy Diversity 6–13/Immunoglobulin Heavy Joining 5) rearrangement, can serve as an independent predictor of RT [52]. This BCR subgroup binds to vimentin, a molecule exposed on microenvironmental cells during apoptosis, and promotes the development, survival, and expansion of leukemic cells [54]. Interestingly, the presence of BCR subgroup 8 is often associated with trisomy 12 and *NOTCH1* mutations in RT [12,55]. *BTK* mutations are often found in patients with progressive CLL or those resistant to targeted therapy, and also in RT. A sequencing study found that 50% of RT patients had *BTK* mutations, with mutation sites similar to those seen in CLL progression or targeted therapy resistance [56]. Interestingly, Splicing Factor 3B Subunit 1 (*SF3B1*) mutations were more frequent than *BTK* mutations in RT (67% vs. 50%), although this finding varies across studies [44,56]. Zeta-Chain-Associated Protein Kinase 70 (ZAP-70), a marker for differentiated T cells and NK cells, plays a role in the transition of pro-B cells to pre-B cells in the bone marrow [57]. The aberrantly high expression of ZAP-70 expression is often linked to aggressive disease and is closely associated with RT [58,59,60]. ZAP-70 enhances Immunoglobulin M (IgM) signaling, chemokine expression, and protein synthesis in high-risk CLL, which may affect cell proliferation and apoptosis, and fuel the CLL-RT transformation process, although the mechanisms remain to be explored [61]. Nuclear Factor of Activated T cells 2 (NFAT2), a major anergic regulatory factor downstream of BCR signaling in CLL, maintains the indolent disease phenotype by regulating survival factors such as CD40 Ligand (CD40L) and B Lymphocyte Stimulator (BLYS). The deletion of *NFAT2* gene in B cells accelerates CLL and resembles RT disease in Eμ-TCL1 transgenic mice, primarily due to the preferential use of certain VDJ recombinations (Variable, Diversity, Joining recombinations) and the selection of unmutated *BCR*, leading to oligoclonal disease [62]. The absence of its downstream target gene Lymphocyte-specific Protein Tyrosine Kinase (LCK) further promotes RT [63].

### 2.6. Other Gene Alterations

In addition to these well-studied genetic variations above, some other gene alterations have been found related to RT with lower frequency. Anna Schuh et al. identified novel recurrent single nucleotide variants (SNVs), small insertions, or deletions in 18 genes, including Dual Specificity Phosphatase 2 (*DUSP2*), Svilin (*SVIL*), Dead End Homolog 1 (*DND1*), Delta Notch-Like EGF Repeats 1 (*DNER*), and Immunoglobulin Superfamily Member 3 (*IGSF3*), through the whole-genome sequencing (WGS) of blood and tissue samples from patients treated with CHOP-O (obinutuzumab) in RT patients [55]. They also discovered genomic abnormalities not previously implicated in RT, such as tumor necrosis factor receptor-associated factor 3 (*TRAF3*) and protein tyrosine phosphatase receptor type D (*PTPRD*). *PTPRD* is a tumor suppressor gene co-localized with *CDKN2A*, which is often silenced in cancers through promoter hypermethylation. In RT patients, *PTPRD* is frequently deleted, inactivated by mutations, or downregulated [55]. BCL6 Corepressor (*Bcor*) loss drives alterations in the B-cell compartment, promotes the transformation of CLL into aggressive lymphoma, and frequently co-occurs with mutated *NOTCH1* which is a hallmark of RT, suggesting the therapeutic potential of targeting *Bcor* and *NOTCH1* in RT [64].

## 3. Gene Instability

Genomic instability, driven by both specific gene alterations and broader molecular mechanisms, represents a critical factor in the pathogenesis of RT. Key contributors include aberrant somatic hypermutation (SHM), the dysregulation of the DNA damage response (DDR) pathway, telomere dysfunction, and complex karyotype (CK). These mechanisms synergistically drive malignant transformation by accumulating oncogenic mutations, promoting clonal evolution, and enhancing cellular stress tolerance, ultimately leading to the aggressive progression and poor clinical outcomes characteristic of RT.

Among these mechanisms, aberrant somatic hypermutation (SHM) plays a significant role in promoting genomic instability in RT [65]. In normal B cells, SHM occurs during the immune response within germinal centers (GCs), where it introduces targeted mutations in immunoglobulin genes to generate high-affinity antibodies [66]. However, aberrant SHM was observed in RT. It was induced by elevated levels of activation-induced cytidine deaminase (AID) expression, which extends SHM beyond Ig genes, affecting key regulatory regions of proto-oncogenes such as *MYC*, Rho Family GTPase H (*RhoH*), Paired Box 5 (*PAX5*), and Proviral Integration site for Moloney Murine Leukemia Virus 1 (*PIM1*) [65]. This off-target SHM contributes to genomic instability by disrupting the normal regulation of these genes, thereby promoting oncogenic transformation and aggressive disease progression.

Aberrations in the DNA damage response (DDR) pathway also contribute to genomic instability in RT. An abnormal gene expression pattern associated with DDR has been observed in RT, including the overexpression of Poly ADP-ribose Polymerase (*PARP*) and Fanconi Anemia Complementation Group Protein G (*FANCG*), coupled with the downregulation of RAD52 Homolog (*RAD52*), RNA Polymerase II Subunit J *(POLR2J*), *Breast Cancer Type 2 Susceptibility Protein (BRCA2*), and Ataxia Telangiectasia and Rad3-related Protein (*ATR*) [55]. Clones harboring DDR pathway mutations exhibit a survival advantage during RT progression, likely due to impaired DNA repair mechanisms that foster clonal evolution and resistance to genotoxic stress [67]. These defects highlight the role of DDR dysregulation in enabling RT cells to tolerate and accumulate genomic alterations [27,62].

Telomere dysfunction is another important contributor to genomic instability in RT. Telomere length ≤ 5000 base pairs has been identified as an independent predictor of RT, highlighting the role of telomere shortening in disease progression [68]. Telomeres are protected by the Shelterin protein complex, which includes Protection of Telomeres 1 (POT1), a critical component that binds directly to telomeric DNA and maintains telomere integrity. Somatic mutations in *POT1* impair its ability to protect telomeres, leading to telomere erosion and chromosomal abnormalities. Interestingly, these defects in telomere maintenance facilitate the acquisition of malignant characteristics in CLL cells, may ultimately drive their transformation into RT [69].

Complex karyotype (CK), defined as the presence of more than three chromosomal abnormalities, is strongly associated with poor prognosis in CLL and RT [27,70,71,72]. CK encompasses both numerical abnormalities (e.g., monosomy and trisomy) and structural abnormalities (e.g., translocations, deletions, and inversions) [70]. In a study investigating the characteristic immunophenotypic features and cytogenetic alterations in patients with RT, seven out of eight patients had CK, further emphasizing its association with high-risk disease [73]. Notably, RT patients without CK demonstrated significantly higher survival rates under the first-line R-EPOCH chemoimmunotherapy, suggesting that CK is a robust marker of adverse outcomes in RT [15].

Following the comprehensive characterization of gene alterations associated with RT, it is important to note that many of the mutations are identified through the sequencing of samples obtained before and after RT diagnosis. However, it remains unclear whether these mutations are newly acquired during the transformation process or were already present as minor subclones during the CLL phase and subsequently selected during RT progression. Resolving this uncertainty is critical for understanding the clonal evolution of RT and identifying early molecular events that drive transformation. Advanced sequencing technologies, such as single-cell sequencing and deep targeted sequencing, combined with subclonal analysis, could provide deeper insights into the temporal dynamics of these mutations. Additionally, the development of reliable in vitro models that faithfully recapitulate the CLL-to-RT transition would enable mechanistic studies to dissect the role of specific mutations in transformation. Such advancements could ultimately facilitate the early detection of high-risk CLL patients and guide timely therapeutic interventions to prevent or delay RT progression.

## 4. Signaling Pathways

While gene alterations have been extensively studied in RT, abnormalities in signaling pathways remain less understood, largely due to the lack of reliable research models. Among these pathways, the BCR signaling pathway plays a central role in the pathogenesis of B-cell malignancies, including CLL [74,75]. The dysregulated activation of the BCR pathway is a key pathogenic mechanism in CLL and a critical driver of RT. Upon activation, BCR signaling propagates through downstream effectors such as the PI3K/AKT, NF-κB, and mTOR pathways, promoting cellular processes including proliferation, survival, and migration (Table 1 and Figure 2).

### 4.1. PI3K/ATK Pathway

The PI3K/AKT signaling pathway plays a central role in the pathogenesis of RT, with increased expression of PI3K and frequent activation of AKT observed in high-risk CLL patients and in over 50% of RT cases [50,77,80]. This pathway regulates a diverse array of downstream effectors that drive cell proliferation, survival, and metabolic reprogramming, contributing to the aggressive phenotype of RT. One critical downstream consequence of AKT activation is the overactivation of the NOTCH1 signaling pathway, which further promotes oncogenic signaling and disease progression. This interplay suggests that the combined inhibition of PI3K/AKT and NOTCH1 may represent a promising therapeutic strategy for high-risk CLL and RT patients [46,47,48]. *PTEN* (Phosphatase and Tensin Homolog) is a well-known tumor suppressor gene, serving as a key negative regulator of AKT activity. RNA-sequencing analyses have confirmed that the synergistic activation of PI3K and MYC is a hallmark of RT, with a negative correlation observed between PTEN expression and the levels of AKT and MYC in both mouse and human RT models [77]. This molecular interplay underscores the importance of the PI3K/AKT pathway in RT pathogenesis and highlights the potential of inhibitors targeting this pathway. Preclinical studies have demonstrated the efficacy of the PI3K inhibitor in suppressing tumor growth in both mouse and human RT models [77]. Additionally, the PI3K/AKT pathway is intricately linked to cellular metabolism through its interaction with Nicotinamide mononucleotide adenylyltransferase (NAMPT), the rate-limiting enzyme in NAD^+^ biosynthesis. NAMPT is involved in a positive feedback loop with BTK/PI3K/AKT signaling, amplifying oncogenic signaling and promoting cell survival. Inhibitors targeting both PI3Kγ/δ and NAMPT have shown synergistic effects in blocking AKT activation and inducing apoptosis in RT patient-derived xenograft (PDX) models, offering a potential dual-targeting strategy for RT treatment [76].

### 4.2. NF-κB Signaling

The NF-κB signaling pathway, a downstream effector of BCR signaling, is more frequently activated in RT than in CLL. The major components of the NF-κB pathway, such as NF-κB1, RELA, NF-κB2, and RELB, along with downstream factors Inhibitor of κB Kinase γ (IκBKγ), Mitogen-Activated Protein Kinase Kinase Kinase 14 (MAP3K14), Conserved Helix-Loop-Helix Ubiquitous Kinase (CHUK), and Inhibitor of κB Kinase β (IκBKγ), broadly exhibit a high expression pattern in malignant RT cells [78]. Targeting NF-κB activation has demonstrated promising therapeutic efficacies in RT preclinical models. A novel NF-κB inhibitor, IT-901, induces significant cell apoptosis in primary RT cells and RT-PDX models by downregulating the expression of the anti-apoptotic protein XIAP and upregulating the pro-apoptotic protein Bim in tumor cells [79]. Interestingly, in line with the essential role of NF-κB signaling in reprograming the microenvironment in CLL disease [86], IT-901 could also inhibit NF-κB signaling in microenvironmental supportive cells, including HS-5 stromal cells and nurse-like cells (NLCs), and disrupt their tumor-promoting phenotype by suppressing the expression of integrin genes in supporting cells [79]. CARD9, an activator of NF-κB signaling, is highly expressed in the RT cell line U-RT1 and in the neoplastic B cells of primary RT tissue specimens. In U-RT1, CARD9 substitutes for CARD11 within the CBM complex (composed of CARD11, MALT1, and BCL10), and its knockdown significantly reduces cell viability, highlighting CARD9 as a potential therapeutic target for RT [80].

### 4.3. MAPK/RAS/ERK Pathway

Gene mutations associated with the MAPK/RAS/ERK pathway are frequently observed in CLL patients with poor prognosis and are strongly associated with a survival advantage during RT [55]. The overexpression of Protein Tyrosine Phosphatase Non-Receptor Type 11 (PTPN11), a positive regulator of the MAPK-RAS-ERK cascade, is a hallmark of RT samples. Concurrently, Kirsten Rat Sarcoma Viral Oncogene Homolog (KRAS) is predominantly overexpressed in RT, while B-Rapidly Accelerated Fibrosarcoma (BRAF) expression is notably reduced [55]. This imbalance suggests a selective pressure for RAS-driven signaling in RT, which may promote cell survival and proliferation while bypassing BRAF-dependent regulatory checkpoints.

### 4.4. mTOR Pathway

The enrichment of the mTOR signaling pathway has been identified in RT patients resistant to BTK inhibitors, suggesting that mTOR activation may serve as a compensatory survival mechanism [81]. mTOR inhibitors, which block the G1 phase of the cell cycle, could counteract this resistance by targeting downstream cell cycle regulators such as CDKN2A/CDKN2B. This strategy is particularly relevant in the context of *TP53* mutations, where conventional cell cycle checkpoints are already compromised [34]. Combining mTOR inhibitors with BTK inhibitors represents a promising approach to overcome therapeutic resistance in RT.

### 4.5. Other Pathways

In CLL, Bcl-2 is a well-established therapeutic target; however, RT cells exhibit reduced apoptotic sensitivity and diminished dependence on Bcl-2 [82]. This shift is accompanied by the downregulation of pro-apoptotic genes, including Harakiri (*HRK*) and Phorbol-12-myristate-13-acetate-induced protein 1 (*PMAIP1*), the latter of which encodes NIP3-like protein X (NOXA). [82]. Intriguingly, structural changes in mitochondrial cristae—specifically, tightly packed cristae that sequester Cytochrome C (Cyt C)—have been identified as an additional mechanism of intrinsic apoptosis resistance in RT [83]. Metabolically, RT is characterized by heightened oxidative phosphorylation (OXPHOS), a feature distinct from the Warburg effect seen in many cancers [39]. The MGA-NME1 axis has emerged as a critical regulator of this metabolic shift [40]. *MGA*, a functional MYC suppressor mutated in 36% of RT cases (versus 3% in CLL), regulates OXPHOS through its target NME1. MGA deletion elevates OXPHOS levels, enhancing energy production and extending survival in RT mouse models. This metabolic reprogramming underscores the adaptability of RT cells to meet the bioenergetic demands of aggressive proliferation.

Benefitting from the understanding of the dysregulated signaling pathways in RT, current research has identified potential therapeutic targets, such as PI3K/AKT inhibitors, NF-κB inhibitors, and dual-targeting strategies combining BTK and mTOR inhibitors for potential RT treatment. However, the preclinical and clinical outcomes of these treatments are dismal, and significant challenges remain. The lack of reliable preclinical models and the heterogeneity of RT limit the ability to fully elucidate the disease mechanisms and test therapeutic combinations. Moreover, resistance to targeted therapies, particularly in patients with *TP53* mutations or prior exposure to BTK inhibitors, underscores the need for more robust and personalized treatment strategies.

## 5. Immune Microenvironment

The immune microenvironment plays a pivotal role in the pathogenesis of RT, contributing to immune evasion, tumor progression, and therapeutic resistance. Emerging evidence highlights multifaceted dysregulation across immune checkpoints, effector cells, cytokine signaling, and chemotactic pathways, collectively shaping an immunosuppressive niche that facilitates the aggressive evolution of RT (Figure 3).

### 5.1. Immune Checkpoint Molecules

RT is characterized by profound alterations in immune checkpoint molecules, including Programmed Death-1 (PD-1), Lymphocyte Activation Gene-3 (LAG3), and T-cell Immunoreceptor with Ig and ITIM domains (TIGIT), which collectively impair anti-tumor immunity [87]. In RT-DLBCL, PD-1/PD-L1 interactions drive immune tolerance through bidirectional signaling: PD-1^+^ tumor B cells engage PD-L1^+^ histiocytes and dendritic cells, while PD-L1 expression on T cells further dampens cytotoxic activity [88,89]. Strikingly, compared to CLL or DLBCL patients, 80% (12/15) of CLL clonally related RT-DLBCL patients exhibit a significantly increased portion of PD-1^+^ B cells, which suggests that the expression of PD-1 could act as a strong clonal relatedness biomarker in RT [89,90]. Mechanistically, the high expression of PD-1 in RT cells could activate pro-survival AKT/mTOR pathways and upregulates the expression of anti-apoptotic proteins (Bcl-2, Mcl-1, and XIAP), thus promoting tumor resilience, under targeted therapies [91]. PD-1 blockade has shown efficacy in RT, particularly in patients with enriched CD8^+^ effector/memory T cells highly expressing transcription factor Zinc Finger Protein 683 (ZNF683) in peripheral blood and bone marrow [92,93]. LAG3, an inhibitory receptor expressed on activated T cells, NK cells, B cells, and plasmacytoid dendritic cells (pDCs), is also overexpressed in RT and correlates with PD-1 and TIGIT, further entrench immunosuppression [87].

### 5.2. Immune Cell Dysfunction

RT patients exhibit reduced clonality of peripheral blood T cells and elevated FOXP3^+^ regulatory T cell (Treg) infiltration [88]. Paradoxically, in contrast to solid tumors where Tregs correlate with immune suppression, FOXP3^+^ Tregs in RT may predict better prognosis or therapeutic response, suggesting its context-dependent roles in hematologic malignancies [94]. The increased infiltration of CD163^+^ tumor-associated macrophages (TAMs) contributes to an immunosuppressive microenvironment in RT, which could further impair anti-tumor immunity [87].

### 5.3. Cytokine and Chemokine Signaling

Interferon Regulatory Factor (IRF) mediates critical responses to interferon gamma (IFN-γ), a key cytokine in anti-tumor immunity. However, RT cells exhibit a blunted response to IFN-γ, characterized by the impaired phosphorylation of STAT1 and STAT3, two transcription factors essential for IFN-γ signaling [77,84]. This defect likely enables RT cells to evade immune surveillance and sustain survival. The broader dysregulation of cytokine signaling pathways, including JAK-STAT, may further compromise immune homeostasis, underscoring the need to restore cytokine responsiveness as a therapeutic strategy. The CXCL12-CXCR4 (C-X-C motif chemokine ligand 12–C-X-C motif chemokine receptor 4) axis, which regulates immune cell homing and retention, is aberrantly activated in RT. In the Eμ-TCL1 mouse model, a CXCR4 mutation (C1013G) drives excessive signaling, leading to cell cycle dysregulation via Polo-like kinase 1 (PLK1)/Forkhead box M1 (FOXM1) activation and recapitulating transcriptional features of RT [85]. This axis may promote tumor cell dissemination and microenvironment remodeling, offering a target for disrupting RT progression.

The aberrant immune microenvironment in RT represents a critical determinant of disease aggressiveness and therapeutic failure. Its complexity—spanning checkpoint dysregulation, effector cell dysfunction, and cytokine signaling collapse—underscores the need for combinatorial approaches that simultaneously target tumor-intrinsic vulnerabilities and immune escape mechanisms. Notably, the immunosuppressive effects of CLL therapies, combined with the intrinsic immune dysfunction in CLL, likely promote EBV-driven RT, suggesting the importance of considering EBV (Epstein–Barr Virus) status in the clinical management of RT, as it may influence treatment strategies and prognosis [95].

## 6. Epigenetic Modifications

Epigenetic modifications, including DNA methylation, histone modifications, and non-coding RNA regulation, play pivotal roles in the pathogenesis of RT. These modifications dynamically alter gene expression patterns, contributing to the dysregulation of key signaling pathways and driving the aggressive transformation of CLL into high-grade lymphoma.

### 6.1. DNA Methylation

DNA methylation is an epigenetic mechanism that modulates gene silencing or activation through the addition of methyl groups to cytosine residues, predominantly at CpG dinucleotides [96,97]. DNA methyltransferases (DNMTs) are enzymes that catalyze DNA methylation reactions by transferring methyl groups from S-adenosylmethionine (SAM) to specific cytosine residues on DNA, thereby establishing and maintaining methylation patterns critical for genomic regulation [97,98]. Recent genome-wide methylation analyses reveal that clonally related RT retains a CLL-like epigenetic imprint, characterized by focal hypermethylation at stem cell maintenance genes (e.g., *OSM*) and cell cycle regulators (e.g., *CDKN2A*), mediated by dysregulated DNMTs [99]. DNMT3A mutations or deletions occur in ~7% of RT cases [100], while DNMT1 overexpression drives the silencing of Polycomb-repressed targets (e.g., *CDKN2A*) [99]. In contrast, clonally unrelated RT exhibits methylation patterns resembling de novo DLBCL, with distinct hypermethylation of tumor suppressor WT1 and immune checkpoint genes. Hypomethylating agents (e.g., 5-azacytidine) reverse the DNMT-mediated repression of tumor suppressors such as *OSM* (Oncostatin M), restoring its anti-proliferative effects in DLBCL cell models [99]. Stereotyped BCR signaling is strongly associated with epigenetic modifications in RT. For instance, the IGLV3-21^R110^ (Immunoglobulin Light Chain Variable region gene segment 3-21 ^R110^) CLL subpopulation exhibits intermediate DNA methylation levels, positioning it epigenetically between naive-like and memory-like CLL subtypes [101]. This subpopulation is characterized by frequent mutations in SF3B1 and ATM, as well as the epigenetic derepression of genes involved in oxidative phosphorylation, cell cycle regulation, mTOR signaling, and Interferon Regulatory Factor (IRF) pathways [53,101,102]. These changes likely contribute to the propensity of subset 8 CLL to undergo RT. Clonally related RT is further distinguished by global DNA hypomethylation, which disrupts the regulation of pathways such as EZH2, Wnt, PI3K/AKT, and IGFR1. These epigenetic alterations are rarely observed in clonally unrelated RT and may underlie the drug resistance and poor prognosis associated with clonally related disease [39,102]. Additionally, the hypermethylation of the hMLH1 promoter, which is often accompanied by microsatellite instability, has been observed in RT-DLBCL, suggesting a role for DNA mismatch repair defects in driving genomic instability during transformation [103].

### 6.2. Histone Modifications and Chromatin Remodeling

Histone modifications (e.g., acetylation, methylation, and phosphorylation) dynamically regulate chromatin compaction by altering the chemical states of histones, thereby governing transcriptional accessibility and functional gene activity [104,105]. Chromatin remodeling, an ATP-dependent process involving nucleosome repositioning or compositional rearrangement, directly controls transcriptional initiation or repression through the reorganization of DNA–histone interactions mediated by complexes such as Chromodomain Helicase DNA-binding proteins (CHD) and Switch/Sucrose Non-Fermentable (SWI/SNF) [106,107].

*EZH2* (a PRC2 subunit catalyzing H3K27me3) is deleted or mutated in 15% of clonally related RT, leading to the derepression of oncogenic MYC and PI3K/AKT pathways [38,39,100]. Protein arginine methyltransferase 5 (PRMT5), an epigenetic modifier, catalyzes the symmetric dimethylation of arginine residues on histones (e.g., H3 and H4) and non-histone proteins [108]. PRMT5 is overexpressed in progressive CLL patients and RT patients, activating the oncogenic pathways to promote CLL progression and aggressive transformation [108]. Similarly, Chromodomain Helicase DNA Binding Protein 2 (CHD2), an ATP-dependent chromatin remodeling enzyme, is associated with transcriptional reprogramming in RT [77]. In RT mouse models with *CHD2* mutations, the expression of its target gene *PTPRN6* is decreased, highlighting the role of chromatin remodeling in RT progression [77]. Furthermore, mutations in chromatin remodelers *SETD2* (H3K36 methyltransferase) and *ARID1A* (a critical subunit of the SWI/SNF chromatin remodeling complex) occur in 9% and 8% of RT cases, respectively, disrupting transcriptional fidelity and DNA damage repair [38,39,100].

### 6.3. Non-Coding RNA Regulation

MicroRNAs (miRNAs) are critical regulators of post-transcriptional gene expression, influencing cell proliferation, differentiation, stress responses, and metabolism [109,110]. In CLL, the most common chromosomal abnormality—the deletion of 13q14—often results in the loss of miR-15a and miR-16-1, which are key tumor suppressors [111]. Epigenetic alterations in RT also involve the dysregulation of miRNAs due to DNA copy number variations, abnormal miRNAs processing, and localization within cancer-related genomic regions [112,113,114]. Specific miRNA signatures have been identified in RT, including the downregulation of miR-146b, miR-21, and miR-181b, and the upregulation of miR-150 [114,115,116]. The functional analysis of these miRNAs reveals enrichment in cancer-related pathways, such as P53 and STAT3 signaling [114]. miR-19b is significantly upregulated in RT and promotes tumorigenesis by upregulating Ki67 and downregulating p53 [117]. The high expression of miR-125a-5p or low expression of miR-34a-5p can predict RT with high specificity, offering potential biomarkers for early detection [118]. miR-17/92 could promote the lymphoproliferative disease by inducing MYC expression, suppressing the expression of tumor suppressor PTEN and the pro-apoptotic protein Bim [119]. The typical strong MYC activity in human and mouse RT, as well as the focal amplification of the miR-17/92 cluster observed in ∼9% of human RT cases, further implicates a role of miR-17/92 in RT [38,119]. miRNAs are also detected in extracellular vesicles, such as exosomes, within the tumor microenvironment. Notably, miR-19b, which is found in exosomes, may serve as a diagnostic marker for RT. These extracellular miRNAs contribute to intercellular communication and may play a role in modulating the immune microenvironment, further promoting RT progression [117].

## 7. Current Advances in Models for Studying Molecular Pathogenesis of Richter’s Transformation

As highlighted in this review, the current understanding of the molecular mechanisms in CLL-RT is largely derived from clinical samples, with limited progress in developing robust preclinical models. The challenges in genetically manipulating B cells and the incomplete understanding of RT mechanisms have hindered the establishment of reliable cell lines and animal models, thereby restricting deeper mechanistic exploration. The development of such models is critical not only for elucidating the molecular drivers of RT but also for providing platforms to test novel therapeutic agents and combination therapies.

### 7.1. Cell Line Models

The first and only recognized RT cell line, U-RT1, was established in 2021 from a cervical lymph node biopsy of a 60-year-old male CLL patient who developed clonally related DLBCL. U-RT1 has been well characterized through cytogenetic, chromosomal, and genetic analyses, as well as immune response assays, confirming its utility as a model system for studying RT pathogenesis and therapeutic approaches [120]. However, the lack of additional RT cell lines limits the ability to capture the heterogeneity of RT.

### 7.2. Animal Models

Most animal models of RT rely on CLL models, with the Eμ-TCL1 transgenic mouse being the most commonly used. These mice overexpress TCL1 in B cells, leading to the development of aggressive lymphomas resembling DLBCL [121,122]. However, TCL1 overexpression is not specific to CLL and can also drive mature T-cell leukemia, limiting its ability to faithfully recapitulate the evolution from indolent CLL to RT [123,124]. To address this, researchers have crossed Eμ-TCL1 mice with constructs driving the constitutive activation of AKT, NOTCH1, MYC, and PRMT5 to study specific molecular pathways in RT [34,50,108,125]. Additionally, loss-of-function approaches, such as CRISPR-Cas9 gene editing, have been employed to model RT. For example, Ten Hacken et al. demonstrated that the co-deletion of multiple CLL drivers (such as *TRP53*, *MGA*, and *CHD2*) in del (13q) background mice drives lymphoma transformation, providing proof of concept for the role of genetic co-operativity in RT [77]. Similarly, Chakraborty et al. evidenced that simultaneously suppressed the expression of *CDKN2A*, *CDKN2B*, and *TP53* in murine leukemic cells from Eμ-TCL1 mice accelerated the cell proliferating and histologically recapitulated RT disease in recipient mice [34]. Iyer et al. identified the MGA-Nme1-OXPHOS axis as a key regulator of RT by observing rapid CLL-to-RT transformation in recipient mice following the deletion of the *MGA* gene in the Sf3b1/Mdr CLL mouse model [40].

Patient-derived xenograft (PDX) models have also emerged as valuable tools for studying RT. In 2018, Silvia Deaglio’s team established two PDX models, RS9737 and RS1316, from RT patients with distinct genetic mutations. RNA sequencing confirmed that over 80% of the transcriptome is shared between the primary tumors and PDX models, highlighting their fidelity for drug discovery and therapeutic testing [126]. Similarly, CD19-expressing RT-DLBCL cells (HPRT3, HPRT2, and HPRT1) derived from core biopsies of RT-DLBCL cases have been used to construct RT-PDX models via tail vein injection, enabling the preclinical evaluation of novel therapies [127].

### 7.3. Challenges and Future Directions

Despite these advances, the development of RT cell lines and animal models remains in its early stages. The U-RT1 cell line, while promising, has not been widely adopted, and the heterogeneity of RT necessitates the establishment of additional cell lines representing diverse molecular subtypes. Similarly, the lack of standardized methods for constructing and evaluating animal models complicates cross-study comparisons and validation. Future efforts should focus on developing more representative cell lines, refining animal models to better mimic the transition from indolent CLL to RT, and leveraging advanced technologies such as CRISPR-Cas9 and multi-omics approaches to uncover novel molecular drivers and therapeutic targets. Addressing these challenges will be essential to advance our understanding of RT and improve outcomes for patients with this aggressive disease.

## 8. Conclusion and Prospects

Despite significant progress in characterizing the genetic and molecular landscape of RT, current understanding remains largely descriptive, focusing on observable phenomena such as gene mutations and signaling pathway abnormalities without fully elucidating the underlying mechanistic drivers. A critical barrier to advancing RT research is the lack of robust preclinical models that accurately recapitulate the complex biology of RT, limiting the ability to dissect its molecular pathogenesis and test novel therapeutic strategies. Consequently, treatment for RT remains largely extrapolated from CLL therapies, with no targeted therapies specifically designed for RT. This is particularly problematic given that most RT patients have already been exposed to targeted inhibitors during their CLL treatment, often leading to therapeutic refractoriness and poor outcomes.

A deeper understanding of the molecular mechanisms driving RT is essential to address these challenges. Elucidating the interplay between genetic, epigenetic, signaling, and microenvironmental factors will not only optimize diagnostic methods, enabling early detection and timely intervention, but also pave the way for personalized treatment strategies tailored to the unique biology of RT. Furthermore, identifying novel molecular targets and vulnerabilities specific to RT could lead to the development of innovative therapies that overcome resistance mechanisms and improve patient outcomes.

Future research should prioritize the development of advanced preclinical models, such as patient-derived xenografts and organoid systems, to better mimic the complexity of RT and facilitate mechanistic studies. Additionally, integrating multi-omics approaches—including genomics, transcriptomics, epigenomics, and proteomics—will be crucial to unravel the molecular heterogeneity of RT and identify actionable therapeutic targets. By focusing on the unique molecular characteristics that distinguish RT from CLL, researchers can uncover new avenues for intervention and ultimately transform the clinical management of this aggressive disease.

## Figures and Tables

**Figure 1 ijms-26-05563-f001:**
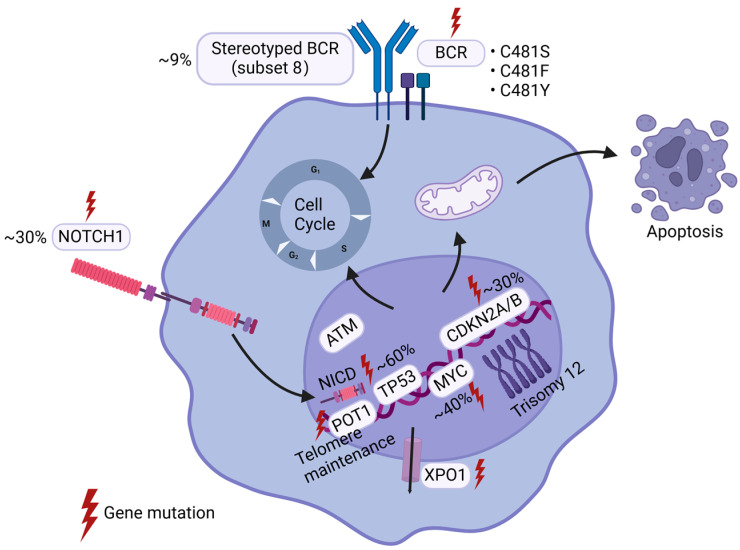
Genetic Alterations of Richter’s Transformation. Richter’s Transformation is molecularly characterized by recurrent genetic alterations in TP53 (~60%), CDKN2A/B (~30%), NOTCH1 (~30%), and MYC (~40%). These mutations frequently co-occur and collectively dysregulate critical cellular processes, including signal transduction, proliferation, cell cycle, and apoptosis. Aberrations in the BCR signal pathway, particularly mutations in BCR components and BTK, remain pivotal in RT pathogenesis. Advances in sequencing technologies continue to unveil novel genetic drivers implicated in this aggressive transformation.

**Figure 2 ijms-26-05563-f002:**
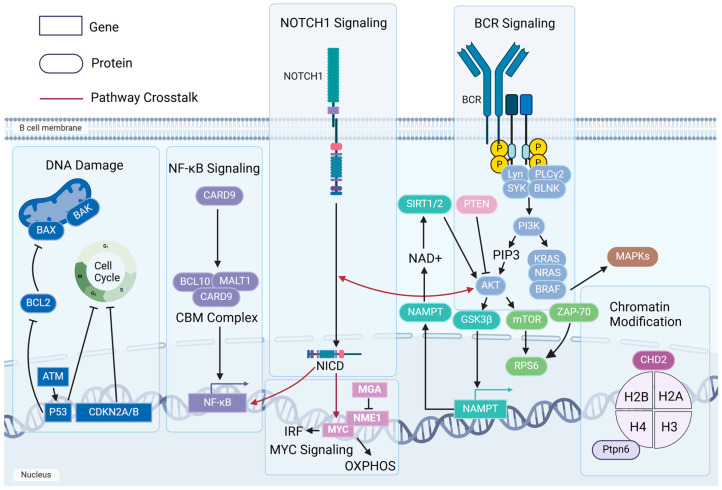
Abnormal signal pathways of Richter’s Transformation. The BCR signaling axis retains its central role in RT, orchestrating malignant progression through downstream effectors such as PI3K/AKT, NF-κB, and mTOR, which drive proliferation, survival, and metabolic reprogramming. The synergistic activation of the PI3K/AKT pathway with NOTCH signaling, frequent mutations in NF-κB pathway components, the dysregulation of the MAPK/RAS/ERK cascade, and compensatory mTOR pathway hyperactivation collectively fuel the aggressive phenotype of RT. Metabolic adaptations, including enhanced OXPHOS, further bolster cellular fitness and treatment resistance.

**Figure 3 ijms-26-05563-f003:**
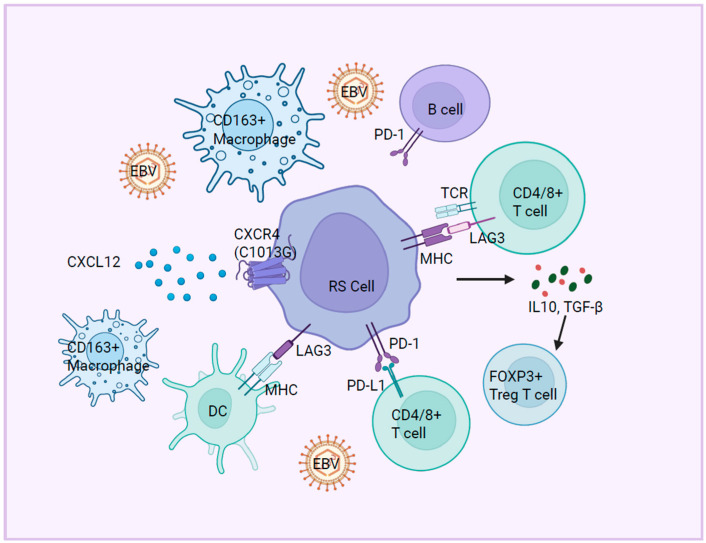
Tumor microenvironment of Richter’s Transformation. The immune microenvironment of RT is marked by the aberrant expression of immune checkpoint molecules (e.g., PD-1 and LAG3), which impair anti-tumor immunity and promote immune evasion. Concurrently, immune dysfunction—manifested by the increased infiltration of regulatory T cells (Tregs) and CD163^+^ tumor-associated macrophages—and defective cytokine signaling exacerbate immune suppression, fostering a permissive niche for disease progression.

**Table 1 ijms-26-05563-t001:** The relationship of signaling pathways and their functions for RT. The table primarily illustrates several key signaling pathways (e.g., PI3K/AKT, NF-κB, MAPK/RAS/ERK, and mTOR) and their corresponding regulatory factors, delineating their specific associations with RT progression, such as cell survival, proliferation, and metabolism, as well as potential therapeutic agents targeting these pathways in RT treatment.

Signaling Pathways	Regulators	Functions
PI3K/AKT	BTK↑ * [76]	Cell survival, proliferation, and metabolic reprogramming
	PI3K↑ [50,77]	Cell survival, proliferation, and metabolic reprogramming
	AKT↑ [50]	Cell survival, proliferation, and metabolic reprogramming
	NOTCH1↑ [46,47,48]	Tumor microenvironment and RT progression
	MYC↑ [77]	Hallmark of RT and RT progression
	PTEN↓ [77]	Key negative regulator of AKT activity
	NAMPT↑ [76]	Cell survival and metabolism
NF-κB	BTK↑	Cell survival, proliferation, and metabolic reprogramming
	Major components of the NF-κB pathway↑ [78,79]	Cell survival, apoptosis, and tumor microenvironment
	CARD9↑ [80]	Cell activity
MAPK/RAS/ERK	PTPN11↑ [55]	Positive regulator of the MAPK-RAS-ERK cascade
	KRAS↑ and B-Raf↓ [55]	A selective pressure for RAS-driven signaling in RT, which may promote cell survival and proliferation
mTOR	Major components of the mTOR pathway↑ [81]	Cell survival, drug resistance, and cell cycle
	TP53↑ [34]	Cell cycle
Others	Bcl-2↑ [82,83]	Cell apoptosis
	MGA↓ [40]	A functional MYC suppressor, which regulates oxidative phosphorylation, metabolic reprogramming, and cell aggressive proliferation
	JAK/STAT↑ [77,84]	Tumor microenvironment
	CXCL12/CXCR4↑ [85]	Tumor microenvironment
	PLK1/FOXM1↑ [85]	Cell cycle and tumor microenvironment

* The upward arrow symbol ↑ denotes the upregulated expression of the molecular factor, while the downward arrow symbol ↓ denotes the downregulated expression of the molecular factor.

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
