# Peer review of "Molecular Features Accompanying Richter’s Transformation in Patients with Chronic Lymphocytic Leukemia"

_ijms, 2025, doi:10.3390/ijms26125563_

Round 1
Reviewer 1 Report
Comments and Suggestions for Authors
The submitted paper tries to summarize the current knowledge about the special clinical situation of CLL transformation into a more aggressive disease, the so-called Richter's transformation. This may be of interest to the readers and therefore I consider it appropriate to publish such review papers. However, for this paper to be accepted, the following comments must be taken into account:
- The title of the article is: "Understanding the Molecular Mechanisms Underlying Richter's Transformation in Chronic Lymphocytic Leukemia". Unfortunately, no understanding of the principles and mechanisms involved in this cell transformation can be gleaned from the article. I admit that this is not possible given the current state of knowledge, so I suggest that the title of the article be changed to "Molecular features accompanying Richter's Transformation in patients with CLL".
- When genes for cyclin-dependent kinase inhibitors are mentioned, it would be useful to also mention the expression of CDKs and cyclins themselves. Otherwise, this information is incomplete for estimating cell cycle progression.
- Some subsections, such as "DNA damage response", "telomere dysfunction", and "complex karyotype", are very brief. They could certainly be expanded to include additional findings from the literature.
- In all subsections of section 4, it would be useful to define the relationship of the described pathways to cell survival, proliferation, entry into apoptosis or autophagy.
- In the description of what is known about the involvement of epigenetic modifications, I found the facts about the expression of DNMTs, genes encoding histone modifications enzymes, etc. to be lacking.
- The paper contains many abbreviations that are not explained in the text. The review article must be written in such a way that it can be easily read by people who are not directly familiar with the field. Therefore, please systematically explain each abbreviation and possibly include a chapter on abbreviations at the end of the paper.
No comments
Author Response
Comment1: The title of the article is: "Understanding the Molecular Mechanisms Underlying Richter's Transformation in Chronic Lymphocytic Leukemia". Unfortunately, no understanding of the principles and mechanisms involved in this cell transformation can be gleaned from the article. I admit that this is not possible given the current state of knowledge, so I suggest that the title of the article be changed to "Molecular features accompanying Richter's Transformation in patients with CLL".
Response1: Thank you for this comment, we feel that is a reasonable suggestion, we have changed the title to “Molecular features accompanying Richter’s Transformation in patients with CLL”.
Comment2: When genes for cyclin-dependent kinase inhibitors are mentioned, it would be useful to also mention the expression of CDKs and cyclins themselves. Otherwise, this information is incomplete for estimating cell cycle progression.
Response2: We have removed the content regarding the CDK2/9 inhibitor (previously in Section 7.1) from our manuscript. After conducting a thorough review of current research, we found insufficient evidence linking CDK or cyclin expression to RT, apart from the well-established CDKN2A/B gene mutations.
Comment3: Some subsections, such as "DNA damage response", "telomere dysfunction", and "complex karyotype", are very brief. They could certainly be expanded to include additional findings from the literature.
Response3: We agree with this comment. As each part of this section contained limited advances, we have consolidated the relevant section by removing subtitles and expanding the explanatory text to improve flow and coherence.
Comment4: In all subsections of section 4, it would be useful to define the relationship of the described pathways to cell survival, proliferation, entry into apoptosis or autophagy.
Response4:We have clarified the relationship between signaling pathways and cellular functions in the text. Additionally, we have created a table summarizing currently investigated signaling pathways and their regulatory roles in RT.
Comment5: In the description of what is known about the involvement of epigenetic modifications, I found the facts about the expression of DNMTs, genes encoding histone modifications enzymes, etc. to be lacking.
Response5: We acknowledge that DNMTs and histone modification genes play important roles in RT. We have significantly expanded this content by: Including DNMT expression patterns in the "DNA methylations" subsection; Adding details about the funtions of EZH2 and SETD2 genes in the "Histone Modifications and Chromatin Remodeling" section.
Comment6: The paper contains many abbreviations that are not explained in the text. The review article must be written in such a way that it can be easily read by people who are not directly familiar with the field. Therefore, please systematically explain each abbreviation and possibly include a chapter on abbreviations at the end of the paper.
Response6: We have thoroughly checked all abbreviations in the manuscript and ensured they are defined at first use. Due to length considerations, we have not included a separate abbreviations section.
Reviewer 2 Report
Comments and Suggestions for Authors
This review is interesting because it addresses a real unmet need for CLL patients. The topic is covered comprehensively, through a complete overview of all the actors involved in RT, supported by adequate bibliographic references. The images are clear and explanatory. The numerous mechanisms and pathways leading to aggressive transformation and the potentially targetable pathways are well illustrated.
I suggest removing the word "however" from line 12 of the abstract.
Some abbreviations should be explained (for example Dll1, line 143).
Adding some tables listing the cited genes, their biological function, their role in transformed disease, and the percentage of mutation or deletion in CLL and radiotherapy could help organize the information and make the article less monotonous and more engaging.
Author Response
This review is interesting because it addresses a real unmet need for CLL patients. The topic is covered comprehensively, through a complete overview of all the actors involved in RT, supported by adequate bibliographic references. The images are clear and explanatory. The numerous mechanisms and pathways leading to aggressive transformation and the potentially targetable pathways are well illustrated.
Comment 1: I suggest removing the word "however" from line 12 of the abstract.
Response 1: We have made this modification and removed the relevant content from the abstract.
Comment 2: Some abbreviations should be explained (for example Dll1, line 143).
Response 2: We thank both reviewers for noting the lack of abbreviation explanations. We have carefully reviewed the manuscript and ensured all abbreviations are properly defined at their first occurrence.
Comment 3: Adding some tables listing the cited genes, their biological function, their role in transformed disease, and the percentage of mutation or deletion in CLL and radiotherapy could help organize the information and make the article less monotonous and more engaging.
Response 3: While Professor Wu et al. (Blood, 2023. 142(1): 11-22) have comprehensively summarized gene mutations and their functions in RT, we have enhanced our manuscript by adding a novel table summarizing abnormal signaling pathways and their downstream regulatory mechanisms in RT, which improves readability and provides additional value.
Round 2
Reviewer 1 Report
Comments and Suggestions for Authors
The authors have taken my recommendation into consideration and I consider their manuscript to be acceptable.